# Assessment of Oxidative Stress-Induced Oral Epithelial Toxicity

**DOI:** 10.3390/biom13081239

**Published:** 2023-08-11

**Authors:** Ali I. Mohammed, Simran Sangha, Huynh Nguyen, Dong Ha Shin, Michelle Pan, Hayoung Park, Michael J. McCullough, Antonio Celentano, Nicola Cirillo

**Affiliations:** 1Melbourne Dental School, The University of Melbourne, Carlton, VIC 3053, Australia; mohammed.a@unimelb.edu.au (A.I.M.); huynhn1@student.unimelb.edu.au (H.N.); m.mccullough@unimelb.edu.au (M.J.M.); antonio.celentano@unimelb.edu.au (A.C.); 2College of Dentistry, The University of Tikrit, Tikrit 34001, Iraq; 3College of Dentistry, University of Jordan, Amman 11942, Jordan

**Keywords:** epithelial injury, reactive oxygen species, oxidative stress, oral keratinocytes

## Abstract

Reactive oxygen species (ROS) are highly reactive molecules generated in living organisms and an excessive production of ROS culminates in oxidative stress and cellular damage. Notably, oxidative stress plays a critical role in the pathogenesis of a number of oral mucosal diseases, including oral mucositis, which remains one of cancer treatments’ most common side effects. We have shown previously that oral keratinocytes are remarkably sensitive to oxidative stress, and this may hinder the development and reproducibility of epithelial cell-based models of oral disease. Here, we examined the oxidative stress signatures that parallel oral toxicity by reproducing the initial events taking place during cancer treatment-induced oral mucositis. We used three oral epithelial cell lines (an immortalized normal human oral keratinocyte cell line, OKF6, and malignant oral keratinocytes, H357 and H400), as well as a mouse model of mucositis. The cells were subjected to increasing oxidative stress by incubation with hydrogen peroxide (H_2_O_2_) at concentrations of 100 μM up to 1200 μM, for up to 24 h, and ROS production and real-time kinetics of oxidative stress were investigated using fluorescent dye-based probes. Cell viability was assessed using a trypan blue exclusion assay, a fluorescence-based live–dead assay, and a fluorometric cytotoxicity assay (FCA), while morphological changes were analyzed by means of a phase-contrast inverted microscope. Static and dynamic real-time detection of the redox changes in keratinocytes showed a time-dependent increase of ROS production during oxidative stress-induced epithelial injury. The survival rates of oral epithelial cells were significantly affected after exposure to oxidative stress in a dose- and cell line-dependent manner. Values of TC50 of 800 μM, 800 μM, and 400 μM were reported for H400 cells (54.21 ± 9.04, *p* < 0.01), H357 cells (53.48 ± 4.01, *p* < 0.01), and OKF6 cells (48.64 ± 3.09, *p* < 0.01), respectively. Oxidative stress markers (MPO and MDA) were also significantly increased in oral tissues in our dual mouse model of chemotherapy-induced mucositis. In summary, we characterized and validated an oxidative stress model in human oral keratinocytes and identified optimal experimental conditions for the study of oxidative stress-induced oral epithelial toxicity.

## 1. Introduction

Chemotherapy and radiotherapy are highly effective treatments for cancer. However, these treatments can also damage healthy tissues [1,2,3,4,5]. Oral mucositis (OM) is a pathological condition characterized clinically by erythema, ulceration and pseudomembrane formation of the oral mucosa, resulting in severe function-altering pain and discomfort that can affect the patient’s nutritional intake and oral hygiene. This condition can worsen in severity up to the point where patients are not able to comply with routine daily oral care [6,7,8]. Despite its severity, frequency, and economic impact, to date, there is no single intervention that effectively prevents or treats OM and most of the current standard therapies for mucositis are predominantly palliative [9,10].

Identifying innovative strategies to treat OM cannot disregard the pathogenic mechanisms underlying the development of this condition [11,12]. It is now thought that mucositis is determined by a sequence of several biological and cellular events that arise in the cells and tissues of the submucosa (endothelium and connective tissue) and culminate in damage and breakdown of the epithelial barrier, resulting in the development of mucosal ulcers [13]. The first step in this cascade of events is the generation of reactive oxygen species (ROS) [13]. At a molecular level, chemotherapy and radiotherapy-induced mucosal injuries are associated with the production of ROS at early pre-clinical stages and sequential activation of oxidative stress pathways [14,15]. Importantly, recent studies suggest that targeting epithelial cells is a promising strategy to treat antineoplastic treatment-induced mucosal injury [16].

It has been shown that excessive production of ROS, such as hydrogen peroxide, superoxide, and hydroxyl radicals, plays an essential role in cell damage and loss of function in many tissues and organs [17]. Hydrogen peroxide (H_2_O_2_) is a highly reactive compound that induces cellular injury. The H_2_O_2_-induced oxidant model has been widely used to study the response of cells to oxidative stress [18]. Oxidative stress is one of the main causative factors for the induction of cell apoptosis [19]. Incubation of cells with potentially harmful molecules, such as H_2_O_2_, induce sub-cytotoxic stress and bring the cells into a senescence-like state, termed stress-induced premature senescence [20,21,22]. This process, however, is cell type-dependent and we have shown previously that ROS have distinctive effects in oral keratinocytes and fibroblasts [22]. Specifically, oxidative stress in fibroblasts induces cellular senescence and this, in turn, leads to the production of ROS. Oral keratinocytes are more sensitive to oxidative stress and undergo cell death in the presence of potent pro-oxidant stimuli, which clinically translates into the mucosal ulceration seen in OM [22]. This remarkable sensitivity of oral keratinocytes to oxidative stress may, therefore, hinder the development of epithelial cell-based models of OM.

Several in vitro models of the oral mucosal barrier have recently been developed, wherein the effects of either chemotherapy agents or ionizing radiation on oral keratinocyte viability and proliferation were examined, to optimize the mucositis-associated epithelial changes observed in vivo [23,24,25,26,27,28]. Although these models are two dimensional (2D) and do not accurately represent what occurs in situ, where the cells live in three dimensions, monolayer cultures of oral keratinocytes have been widely used to mimic specific parts of the oral mucosal epithelial barrier [23,29].

Taking into account the pro-oxidant mechanisms involved in the pathogenesis of oral mucositis, it seems clear that new research models of oxidative stress-induced mucositis, in which the new therapeutic interventions can be tested in oral mucosal cells, are needed. Several attempts to characterize oxidative stress using in vitro models of oral mucosal inflammation have been conducted recently [22,30,31,32,33,34]. In particular, it is essential to establish a dose–response curve of the effects of ROS in oral keratinocytes. Therefore, the aim of the present study was to establish an in vitro model of oxidative stress-induced oral mucosal toxicity, using hydrogen peroxide and monolayer cultures of oral keratinocytes to simulate what is thought to occur in oral mucositis.

## 2. Materials and Methods

### 2.1. Cell Lines and Culture Conditions

To establish an in vitro model of oxidative stress-induced oral mucosal toxicity with hydrogen peroxide, we used three adherent monolayer cultures of oral keratinocytes (OSCC cell lines H400 and H357, and immortalized normal human oral keratinocytes, OKF6). The cell lines, H400 [35], a human oral alveolar squamous cell carcinoma cell line (Cat#06092006), and H357 [35], a human tongue squamous cell carcinoma cell line (Cat#06092004), were purchased from Sigma-Aldrich, Castle Hill, NSW, Australia. We used the human oral-derived squamous cell carcinoma cell line in our model because the cells are easy to culture and the results have high reproducibility, in analogy with the existing oral mucosa models [23]. The OKF6 [36], immortalized normal human floor of the mouth keratinocyte, was kindly provided by the Oral Health Cooperative Research Centre (OHCRC), The University of Melbourne, Australia. The choice of this cell line was based on work from previous research, which showed that the OKF6/TERT cells are a valuable and reproducible model of normal oral epithelial cells, as they resemble primary oral keratinocytes in studies of cytotoxicity or inducible cytokine and beta-defensin expression [37,38,39,40].

Details of culture conditions for the keratinocyte cell lines have been documented previously [41,42,43]. Cells were grown in 100 mm Petri plastic dishes (Corning 430167, Sigma-Aldrich, Castle Hill, NSW, Australia). The H400 and H357 cell lines were grown in complete growth medium consisting of Dulbecco’s modified Eagle’s medium (DMEM) (D5796) and nutrient mixture F-12 Ham (N6658) in a 1:1 ratio (Sigma-Aldrich, Australia), supplemented with 10% *v*/*v* foetal bovine serum (FBS) (SFBS-F, Bovogen, Keilor East, VIC, Australia), 1% penicillin streptomycin mixture (P4333, Sigma-Aldrich, Castle Hill, NSW, Australia), and 0.5 g/mL hydrocortisone (HC) (H6909, Sigma-Aldrich, Castle Hill, NSW, Australia), in a humidified atmosphere at standard conditions (5% CO_2_, 37 °C). The OKF-6 cells were also cultured in 100 mm Petri plastic dishes (Corning 430167, Corning, NY, USA), using keratinocyte serum-free medium (K-SFM) (#17005-042, Thermo Fisher Scientific, Scoresby, VIC, Australia) containing 25 μg/mL bovine pituitary extract and 0.2 ng/mL human recombinant epidermal growth factor (as per the manufacturer’s instructions), 0.4 mM CaCl_2_, and 1% penicillin streptomycin mixture (P4333, Sigma-Aldrich, Castle Hill, NSW, Australia), and supplemented with 1% *v*/*v* Newborn Calf Serum (NCS) (N4637, Sigma-Aldrich, Castle Hill, NSW, Australia). The OKF6 cells were grown under the same standard conditions (humidified atmosphere, 5% CO_2_ at 37 °C). The keratinocyte cells, when grown to 80% confluency, were, subsequently, detached via a pre-treatment of 10 mM EDTA for 5 min, followed by incubation with 0.25% trypsin in a 1 mM EDTA solution (T4049, Sigma-Aldrich, Castle Hill, NSW, Australia), for 3 min. The viability of the keratinocytes was confirmed by trypan blue exclusion (trypan blue dye, 0.4% solution, 1450021, Bio-Rad, Hercules, CA, USA).

### 2.2. Optimizing Seeding Density of Cells

In preliminary experiments, the optimization of the human oral keratinocyte cell number for FDA fluorescent assay in 96-well plates was tested in a cell proliferation assay using monolayer cultures of oral keratinocytes (H400, H357, and OKF6). Briefly, H400, H357, and OKF6 cells were seeded in 96-well plates at four different seeding densities (5 × 10^3^, 1 × 10^4^, 1.5 × 10^4^, and 2 × 10^4^ cells/well). Cells were then incubated overnight to allow cell adhesion to the plate, and, subsequently, studied over 72 h. The culture medium was replenished every 24 h. Fluorescent reading was performed and cell proliferation, as a function of fluorescence, was measured every 24 h at four different time points (day-0, day-1, day-2, and day-3). Data were expressed as mean cell proliferation attributed to the FDA fluorescent intensity. Seeding density that remained in the exponential phase, from day-0 up until day-3, was chosen as a standardized seeding density for each cell line. Cell confluence over time was measured by inverted microscopy imaging (FLoid™ Cell Imaging Station, Cat#: 4471136, Life Technologies Australia Pty Ltd., Mulgrave, VIC, Australia).

### 2.3. Induction of Oxidative Stress by H_2_O_2_ Treatment

A series of preliminary experiments to establish an in vitro oxidative stress-induced oral mucositis model by using H_2_O_2,_ a well-known inducer of oxidative stress [18,44], was performed. The effect of H_2_O_2_ dose and duration, inducing cellular injuries, without completely compromising cell culture viability were tested in cell-based assays using monolayer cultures of oral mucosa keratinocytes (H400, H357, and OKF6 cell lines). Briefly, following overnight incubation after seeding, cells were exposed to oxidative stress for up to 24 h. Various concentrations of H_2_O_2_, ranging between 100 and 1200 μM diluted in culture media, were tested over the given time points. The initial range of concentrations was determined based on existing literature [22,45,46,47]. At the end of the incubation period the morphological changes were assessed and photographed with a FLoid cell imaging system (FLoid™ Cell Imaging Station, Life Technologies Australia) using white light and 20× magnification. Subsequent to treatment with H_2_O_2_, the cells’ viability/proliferation were measured by using a trypan blue exclusion assay, fluorescence-based live–dead assay, and fluorometric cytotoxicity assay (FCA) to determine the optimal time and concentration of H_2_O_2_ to induce cellular injuries.

#### 2.3.1. Cell Viability and Growth Assay

In a preliminary experimental phase, the effect of different H_2_O_2_ doses and duration on cell viability and proliferation were assessed using monolayer H400 cultures. Briefly, keratinocytes cells were seeded into 6-well plates (1 × 10^5^ cells/well) and grown in a humidified atmosphere at standard conditions (5% CO_2_, 37 °C). At 60% confluence, cells were exposed to increasing concentrations of H_2_O_2_ (0, 200, 400, and 800 μM) diluted in 1.5 mL of experimental medium and incubated at 37 °C, for up to 24 h. To determine the H_2_O_2_-mediated cell death, the cell number and cell viability in three-wells of each group were counted chronologically at 2, 6 and 24 h timepoints, by using a trypan blue exclusion assay (TC10™ Automated Cell Counter, Bio-Rad, Watford, UK). The viability of control cells (without treatment) was considered 100%, and H_2_O_2_-mediated cell death was assessed by comparing the viability of treated cells with that of control cells. The IC_50_ value was defined as the concentrations of H_2_O_2_ that inhibited the cell viability of treated cells to 50% compared to the untreated control. This preliminary study aimed to assess the best time and the optimal concentration of H_2_O_2_ to induce cellular injury.

#### 2.3.2. Evaluation of Epithelial Cell Viability Rate with Fluorescence-Based Live–Dead Assay

The selected H_2_O_2_ dosage range and duration, as determined by preliminary experiments described above, were further confirmed by assessing the effect of H_2_O_2_ on cell viability and proliferation in monolayer H400 cultures. Briefly, cells were incubated for 24 h with increasing concentrations of H_2_O_2_ (400, 800, and 1200 μM) by using the fluorescence-based live–dead assay. At 24 h after H_2_O_2_ treatment, the H400 cells were labeled with fluorescein diacetate (FDA) and propidium iodide (PI) stains. Two fluorescent dyes allowed two-color discrimination of the population of living cells population from the population of dead cells, following the manufacturer’s instructions. All experiments in this phase were performed in technical triplicates.

#### 2.3.3. Evaluation of Epithelial Cell Viability Rate with Fluorometric Cytotoxicity Assay (FCA)

Normal (OKF6) and malignant (H400 and H357) oral keratinocytes were treated with increasing concentrations of H_2_O_2_ (100 μM–1200 μM) for up to 24 h. Cell viability/proliferation was assessed using a fluorometric cytotoxicity assay (FCA) to determine the H_2_O_2_-mediated cell death. Viability of control cells (without treatment) was considered 100%, and H_2_O_2_-mediated cell death was assessed by comparing the viability of treated cells with that of control cells. The IC_50_ value was defined as the concentrations of H_2_O_2_ that inhibited cell viability of treated cells to 50% compared to untreated control. There were 3–5 replicates used for each group/cell line.

### 2.4. Determination of Intracellular ROS Production and Kinetics

Intracellular ROS production was measured in OKF6 cells by using the cell-permeant 5-(and-6)-chloromethyl-2′,7′-dichlorodihydrofluorescein diacetate (CM-H_2_DCFDA, Invitrogen, Thermo Fisher Scientific, C6827), as per the manufacturer’s instructions. In brief, 1.5 × 10^4^ cells/well were seeded in a dark, clear-bottom 96-well microplate (Sigma-Aldrich) in complete culture media and incubated overnight under standard conditions (5% CO_2_, 37 °C). The media were then replaced and the cells were incubated under standard conditions and allowed to grow until they reached 70–80% confluence. The culture medium was then replaced and OKF6 cells were exposed to 5 µM of CM-H_2_DCFDA solution prepared in Hanks’ balanced salt solution (HBSS) without phenol red (Sigma-Aldrich) and incubated for 30 min in the dark in a humidified incubator under standard conditions. The medium with CM-H_2_DCFDA was then removed, and the plate was washed with HBSS before the addition of growth medium with or without H_2_O_2_ (400 µM) to the wells. CM-H_2_DCFDA is a chloromethyl derivative of H_2_DCFDA, and it is a fluorogenic dye and a useful indicator of hydroxyl, peroxyl, and other ROS activity within a cell. It is a cell-permeable non-fluorescent probe, and after diffusion into the cells, CM-H_2_DCFDA is deacetylated by cellular esterases to a non-fluorescent compound, 2′,7′-dichlorodihydrofluorescein (CM-H_2_DCF), which is later oxidized by ROS into 5-(and-6)-chloromethyl-2′,7′-dichlorofluorescein (CM-DCF), a highly fluorescent compound, staining the cell cytoplasm with bright green fluorescence [48,49]. To determine a time course for intracellular ROS generation, the kinetic increases in fluorescence of CM-DCF were measured at time points 0, 15, 30, and 45 min, and at 30 min intervals thereafter for 6 h at the respective excitation and emission wavelengths of 485/20 nm and 528/20 nm, using the Synergy HTX Multi-Mode Reader (Bio-Tek, Shoreline, WA, USA). ROS generation of the control vehicle was considered (1 unit), and ROS formation was measured as the change in CM-DCF fluorescence (-fold increase) relative to the control vehicle.

### 2.5. Real-Time Quantification of Oxidative Stress in Oral Keratinocytes Using Fluorescent Dye-Based Redox Probes

In further experiments, the real-time measurement of intracellular ROS production at a time point of 3 h was performed using the fluorescent dye-based ROS probes CM-H_2_DCFDA, as per the manufacturers’ instructions. Intracellular ROS measurement was based on the procedure reported by Ro et al. [49], with some modifications. In brief, OKF6 cells were seeded in a cell culture-treated 6-well plate (Sigma-Aldrich) at a density of 0.3 × 10^6^ cells/well in complete culture media and incubated in a humidified atmosphere at standard conditions (5% CO_2_, 37 °C). The cells were allowed to grow until they reached 70–80% confluence, with the growth medium replenished every 48–72 h. At 80% confluence, the cells were pre-treated with 400 µM of H_2_O_2_ and incubated at standard conditions for 4 h. After cell culture treatments, the plate was washed twice with 2 mL/well of PBS, and the media were replaced and OKF6 cells were incubated with 5 µM of CM-H_2_DCFDA solution prepared in Hanks’ balanced salt solution (HBSS) without phenol red (Sigma-Aldrich) and incubated for 30 min in the dark in a humidified incubator under standard conditions. After that, the plate was washed twice with 2 mL/well of HBSS, and the cells were covered with 2 mL/well of HBSS/Ca/Mg and were examined under a FLoidTM cell imaging station digital inverted microscope (ThermoFisher Scientific, Waltham, MA USA) using a filter set for fluorescein (FITC) cell imaging (488 nm excitation and 530 nm emission). The fluorescence intensity excited at 488/530 nm was used as a surrogate indicator of the cell redox (ROS production) status, with an expected increase of the intensity in the presence of oxidative stress. The intensity of fluorescence was measured in seven random and different fields per well per condition in three independent experiments. The images were transferred as “.tiff” files into a computer and were analyzed using the ImageJ software [50,51]. The ROS level of control vehicle was normalized to 1 unit, and the changes in CM-DCF fluorescence were expressed as fold increase relative to the control vehicle.

### 2.6. Immunohistochemistry Analysis

For the in vivo arm of the study, samples from an existing murine model of chemotherapy-induced mucositis [52] were collected. Briefly, C57BL/6 female mice aged 6–12 weeks and weighing between 15 and 20 g received IV saline every 48 h, from day 1 to 13 (control), whereas mice from the treatment group were exposed to IV 5-Flurouracil (5-FU) (50 mg/kg/day), every 48 h, from day 1 to 13, resulting in the development of mucositis [52]. On day 14, the animals were sacrificed, and tongue samples were excised at the tracheal level, harvested, and fixed in 10% buffered formalin for 24 h, at room temperature, in preparation for histological analyses. These tissues were paraffin embedded, sectioned and de-paraffinized, and underwent heat-induced epitope-retrieval (HIER) with 0.1 M citrate buffer. Non-specific epitopes were blocked using the Novocastra Peroxidase Block. For immunohistochemical staining, we used tissues from 6 animals (3 controls and 3 treated) following a protocol reported by us previously [52]. Anti-Myeloperoxidase (ab65871, Abcam, Thebarton, Australia) rabbit polyclonal antibody and the mouse and rabbit specific HRP/DAB IHC detection kit—micro-polymer (ab236466, Abcam, Australia), with goat anti-rabbit IgG secondary antibody (Abcam, Australia) as a secondary antibody, was used. Counterstaining was undertaken with haematoxylin. This methodology was adapted for the use of anti-malondialdehyde antibody (ab27642, Abcam, Australia). Optimization was undertaken to determine the ideal concentration of the primary antibody, which was determined to be 0.2 μg/mL for both MDA and MPO.

Following immunohistochemical staining, the cells within the oral epithelium of the samples were identified via automation, using QuPath open-source digital software v. 0.2.01 “cell positive detection” function, and verified manually via visual assessment. An unpaired t-test, undertaken through the statistical analysis function in GraphPad prism version 8.0.1 for windows (GraphPad Software Inc., San Diego, CA, USA), was used to analyze the differences in positive cell count between the control and 5-FU-treated mice.

### 2.7. Statistical Analyses

Statistical analysis was performed using GraphPad Prism version 8.0.1 for windows (GraphPad Software Inc., San Diego, CA, USA). Cell viability data were expressed as mean percentage viability relative to the untreated control (100%) ± standard deviation (SD). ROS levels are reported as the mean CM-DCF fluorescence change (-fold increase) ± SD relative to control as Control = 1. All other data are presented as the mean ± standard deviation (SD). Statistical significance between two groups was determined by using an unpaired t-test. Statistical significance between two independent groups with more than one dependent variable was compared using multiple t-tests and determined using the Holm–Sidak method. If more than two groups were being compared statistical analysis was conducted using one-way analysis of variance (ANOVA) followed by Tukey’s post hoc test to detect inter-group differences. Multiple linear regressions were evaluated using the regression tool to identify collinearity among the independent variables. Differences were considered statistically significant if *p* < 0.05. The letter “n” indicates the number of independent experiments.

## 3. Results

### 3.1. Growth Kinetics and Optimization of Cell Number for Cell Viability Assay

To identify the optimal seeding density of oral keratinocytes cells, using FDA fluorescent assay, a cell number titration assay was conducted in 96-well plates using three cell lines (H400, H357, and OKF6). Fluorescent intensity over time and experimental replicates for each seeding density per cell line were plotted using GraphPad Prism software to visualize exponential growth. On average, a cell seeding density of 2 × 10^4^ cells/well showed the highest fluorescence signal over time in all the investigated cell lines. In contrast, a seeding density of 5 × 10^3^ cells/well showed the lowest fluorescence signal during all investigated time points (time 0, 24, 48, and 72 h; Figure 1a). We observed steadier cell proliferation when the initial cell seeding density was increased. Furthermore, the time period during which cells were able to grow exponentially also increased (Figure 1a). We also noticed that the increase in the initial cell seeding density ensured that cell confluence remained below ~90% throughout the assay, or that this level of confluence was reached as late as possible in the assay, and this trend was cell-line specific.

We observed that, when cells were seeded at a density of 2 × 10^4^ cells/well, full confluence was reached at the 72 h timepoint (Figure 1c). However, within the time point range 48–72 h, the exact time when the cells became fully confluent was not assessed. Interestingly, despite the presence of mitotic activity, when cells were seeded at 5 × 10^3^ cells/well, the cell confluence remained below ~40% throughout the assay (Figure 1c). In the case of the H400 cell line, the cells showed exponential growth between 0–72 h when seeded at densities of 1 × 10^4^, 1.5 × 10^4^, and 2 × 10^4^ cells/well. Nevertheless, H400 cells seeded at 5 × 10^3^ cells/well showed exponential growth until 48 h, with the growth curve reduced thereafter (Figure 1a). In regard to H357 and OKF6, both cell lines grew exponentially between 0–72 h when seeded at 1.5 × 10^4^ and 2 × 10^4^ cells/well. However, when seeded at 5 × 10^3^ and 1 × 10^4^ cells/well, both H357 and OKF6 cells showed exponential growth until they plateaued at 48 h, after which they showed a growth decrease (Figure 1a).

Subsequently, we investigated the correlation between viable oral keratinocyte cell number and fluorescence intensity over time (Figure 1b). This correlation proved to be linear for cells seeded at a density of 2 × 10^4^ cells/well with correlation coefficients of R = 0.9570, 0.9471, and 0.9685 for H400, H357, and OKF6 cell lines, respectively. A slight decrease of linearity was observed at cell concentrations below 2 × 10^4^ cells/well at an initial seeding density of 1.5 × 10^4^, R = 0.9433, 0.9393, and 0.9384 for the H400, H357, and OKF6 cell lines, respectively. At an initial seeding density of 1 × 10^4^, the correlation coefficients were R = 0.9507, 0.845, and 0.8127 for the H400, H357, and OKF6 cell lines, respectively. The strength of linearity reached the lowest value at a cell concentration of 5 × 10^3^ (R = 0.7396, 0.2297, and 0.4800 for H400, H357, and OKF6 cell lines, respectively (Figure 1b).

Based on the current findings, we concluded that the seeding density should be chosen according to individual cell line growth kinetics in order to ensure the following: (1) cell growth following an exponential pattern after seeding, (2) confluence reached as late as possible, and (3) a reliable linear response observed during the incubation time. In our case, we found that seeding densities of 1–2 × 10^4^ cells/well for H400, 1.5–2 × 10^4^ cells/well for H357, and 1.5–2 × 10^4^ cells/well for OKF6 fulfilled these criteria.

### 3.2. Epithelial Cell Proliferation inhibited by H_2_O_2_ in a Time- and Dose-Dependent Manner

In a series of preliminary experiments designed to establish an in vitro model of oxidative stress-induced oral mucositis by using hydrogen peroxide (H_2_O_2_), we initially used monolayer epithelial cultures (H400 cells) to establish a dose range and duration of H_2_O_2_ treatment that would inhibit cell proliferation and induce cell injury, without completely compromising cell culture viability (Figure 2a). The H-400 cells were incubated for 2 h, 6 h, and 24 h with three different concentrations of H_2_O_2_ (200 μM, 400 μM, and 800 μM), and, then, each of the important factors, such as time of incubation and dose of H_2_O_2_, were validated by determining the best time and the optimal concentration of H_2_O_2_ to induce cellular injury. As reflected by a trypan blue exclusion assay, the results showed a decrease in cell viability after incubation with H_2_O_2_, and this correlated with treatment time and dose. Compared with the untreated control group, H400 cells treated with 200 μM, 400 μM, and 800 μM H_2_O_2_ showed decreasing trends in cell viability after 6 h, and 24 h, but not after 2 h. In contrast, after 2 h exposure to increasing H_2_O_2_ concentrations (200, 400 and 800 μM), H400 cell viability slightly increased. However, this increase was not significant as compared to control (*p* > 0.05; Figure 1a). A significant decrease in cell survival was noted after 24 h incubation with 400 μM and 800 μM H_2_O_2,_ as compared to the control (62.99% ± 8.32, *p* < 0.005, and 47.13% ± 4.88, *p* < 0.001, respectively; Figure 2a). We also investigated the effect of H_2_O_2_-mediated oxidative stress on H400 proliferation (Figure 2b). A progressive increase in the cell number was observed in controls and in the 200 µM H_2_O_2-_treated samples over time. Conversely, treatment with 400 and 800 µM H_2_O_2_ induced a biphasic, time-dependent effect in cell proliferation. In fact, at these concentrations, H400 cell treatment showed an initial increase in proliferation until 6 h after treatment, while at 24 h this trend inverted, with a significant decrease in cell number when compared to the control (*p* < 0.05) (Figure 2b). The pattern of an initial increase in proliferation followed by a subsequent decrease in cell number (biphasic effects on cell proliferation) is a well-documented phenomenon in the field of cellular biology and toxicology [53,54,55,56,57]. Furthermore, assessment of the correlation between total number of cells number and H_2_O_2_ dose over time, a strong negative correlation between total cells number and H_2_O_2_ dose at the 24 h timepoint was observed. This correlation proved to be statistically significant with a linear relationship and a correlation coefficient of R = −0.9277 (*p* < 0.001) (Figure 2c).

Based on our results, we concluded that exposure of H400 cells to H_2_O_2_-induced cellular damage by oxidative stress and caused a decrease in cell viability and proliferation in a time- and concentration-dependent manner. We identified 800 μM as the concentration of H_2_O_2_ that, at 24 h, was able to inhibit H400 cell viability to a proximate level of the IC_50_. Hence, a range of concentrations close to 800 μM and the 24 h timepoint were further explored in subsequent experiments, establishing the oxidative stress-induced injury model in H400 cells.

### 3.3. Cytotoxic Effect of H_2_O_2_ on Oral Keratinocytes

To overcome the limitations due to the use of a viability assay that utilizes the exclusion of certain dyes by live cell membranes (trypan blue exclusion), we used Fluorescein diacetate (FDA) staining that identifies live cells rather than dead cells, as it stains only living cells. While FDA selectively stains viable cells, injured and dead cells were stained with propidium iodide (PI), an analog of ethidium bromide (EB). Therefore, in a subsequent experiment, we tried to validate the results obtained from previous experiments using a simultaneous double-staining procedure with FDA-PI stains.

We found that the identification of cells stained by FDA was unequivocal, with live cells being stained bright green and nonviable cells stained bright red when investigated under an inverted fluorescence microscope with the filter set on the FITC and Texas Red fields for the FDA and PI detection, respectively (Figure 3a). Subsequently, the oxidative effect of H_2_O_2_ was evaluated (Figure 3b,c). Using FDA fluorescence, we demonstrated that exposure of H400 cells to 800 and 1200 μM H_2_O_2_ for 24 h induced a dose-dependent decrease in cell viability of 1200 (Figure 3b). In particular, incubation with 800 μM H_2_O_2_ for 24 h decreased cell viability by about 50% when compared to control cells (43.1% ± 8.45, *p* < 0.05). Furthermore, a strong negative correlation between the cell viability ratio and H_2_O_2_ dose after 24 h of H_2_O_2_ incubation was noticed (R = −0.9033, *p* < 0.001; Figure 3c). These findings were consistent with our previous findings when using a trypan blue assay.

To conclude, we found that the FDA-PI staining is a rapid, cost-efficient, and more reliable method to discriminate live cells when compared to trypan blue exclusion. An FDA-PI assay was, therefore, chosen for subsequent experiments investigating the H_2_O_2_-mediated toxicity in other cell lines.

### 3.4. Establishment of an Oxidative Stress-Induced Mucosal Injury Model with H_2_O_2_-Treated Oral Keratinocytes

In a previous series of preliminary experiments, we determined the optimal experimental conditions, including cell seeding densities, a range of concentrations for H_2_O_2_, and the incubation time to establish an oxidative stress-induced model in monolayer cultures of oral keratinocytes. In a subsequent series of experiments, we established an H_2_O_2_-induced oxidative stress model using three monolayer cultures of oral mucosal keratinocyte, derived from cancer (H400, and H357), and normal (OKF6) oral epithelial cells to simulate what is thought to occur in radiation- or chemotherapy-induced oral mucositis. Based on the findings from the preliminary experiments, oral epithelial cells were incubated, with a broad range of increasing concentrations of H_2_O_2_ (100–1200 μM), for 24 h. The in vitro effects of H_2_O_2_ on H400, H357, and OKF6 cells’ growth and viability were examined after 24 h using a fluorometric cytotoxicity assay (FCA) by using FDA satin. Untreated cells were used as control and the fluorescence density was in proportion to the number of viable cells. The results showed that administration of H_2_O_2_ clearly decreased cell viability in H400 cells after 24 h in a concentration-dependent manner over the examined ranges (200–1200 μM H_2_O_2_). Loss of cell viability after 24 h exposure to H_2_O_2_ was significantly lower in H400 cells compared to untreated control cells for H_2_O_2_ concentrations of 600 μM and higher (Figure 4a). Similarly, the administration of increasing concentrations of H_2_O_2_ (ranging from 100–1300 μM H_2_O_2_) was able to markedly reduce cell viability after 24 h in both H357 and OKF6 cell lines. This effect was significantly different compared to control untreated cells for concentrations of 400 μM and higher, and 100 μM and higher for H357 and OKF6 cells, respectively (Figure 4b,c).

Similarly, administration of increasing concentrations of H_2_O_2_ (ranging from 100–1200 μM H_2_O_2_) markedly reduced cell viability after 24 h in both H357 and OKF6 cell lines. This effect was significantly different compared to control untreated cells for concentrations higher than 400, and 100 μM for H357 and OKF6 cells, respectively (Figure 4b,c).

The calculated IC_50_ values following 24 h exposure to H_2_O_2_ were approximately 800, 800 and 400 μM for H400, H357, and OKF6 cells, respectively (Figure 4b,c). A concentration close to the IC_50_ values for the three cell types (800 μM for H400 cells; 800 μM for H357 cells; 800 μM for OKF6 cells) were considered to be the most appropriate concentrations of H_2_O_2_ to induce cellular injuries in oral epithelial cells and were chosen for successive experiments to investigate H_2_O_2_-mediated toxicity.

### 3.5. Effect of H_2_O_2_-Induced Oxidative Stress on Morphological Changes in Oral Keratinocytes

To evaluate the effect of H_2_O_2_ on cell-morphology, H400, H357, and OKF6 cells were treated with a broad range of concentrations of H_2_O_2_ (100–1200 μM) for up to 24 h. Cell morphology was assessed in bright field micrographs (Figure 5). Morphological observation of untreated cells showed that the cells retained their original polygonal morphology and were stably adherent to the culture plates. In contrast, cells treated for 24 h with H_2_O_2_ at concentrations ≤ 600, 400, and 100 μM, for H400, H357, and OKF6 cells, respectively, displayed morphological alterations compatible with apoptotic cell death.

These features included cell shrinkage, blebbing, condensation and fragmentation of nuclear chromatin, and formation of vesicles resembling apoptotic bodies. Affected cells also displayed other types of morphological alterations, such as loss of polygonal shape at the expense of a round appearance, loss of adhesion with neighboring cells, tendency to remain as single cells rather than grouping in cell colonies, and loss of adhesion on the culture plates (Appendix A).

### 3.6. Assessment of ROS Production during Oxidative Stress-Induced Epithelial Injury In Vitro

Following the optimization of culture conditions, we wanted to confirm if ROS production occurred in keratinocytes treated with H_2_O_2_; thus, mimicking the physiopathology of OM in tissues. To monitor the changes in ROS production in keratinocytes upon H_2_O_2_ treatment, we employed a fluorescent dye-based probe, based on the redox-active green fluorescent (CM-H_2_DCFDA), a highly sensitive thiol peroxidase that is oxidized by H_2_O_2_, yielding a fluorescent adduct that is trapped intracellularly [49]. By introducing the CM-H_2_DCFDA probe in keratinocytes, we could monitor any change in the intracellular redox state in the absence, or in the presence, of H_2_O_2_ in OKF6 cells over a 6 hour time period.

We found that the addition of H_2_O_2_ determined a rapid increase in ROS levels in the experimental group. The increase in ROS levels was seen from the first few minutes after adding H_2_O_2_ and continued throughout all measured time points over the 6 h monitoring time (Figure 5). More precisely, a significant increase (*p* < 0.0001) in the ROS levels in the H_2_O_2_ (400 μM) treated group, compared with that in the control group, started 15 min after the addition of H_2_O_2_ and continued during all the tested time points over the 6 h monitoring period.

The results also showed two time periods when ROS production levels in the H_2_O_2_-treated group plateaued for a while before starting to change again (Figure 5). ROS levels first plateaued (Time plateau-1) between 0.5–1 h and the ROS levels in the H_2_O_2_ treatment group plateaued at about 4.65 ± 0.58 (folds increase), compared to the control group. After that, the ROS production levels started to increase again until they plateaued (Time plateau-2) between 3.5–5.5 h, at about 7.61 ± 0.75 (folds increase), compared to the control group. Time plateau-2 was when H_2_O_2_-induced ROS production reached the highest level. More precisely, 4 h was the time point when ROS production in cells treated with 400 µM H_2_O_2_ reached the maximum level, as determined by the CM-H_2_DCFDA staining kinetic assay (Figure 5).

### 3.7. Dynamic Real-Time Detection of the H_2_O_2_-induced Redox Changes in Oral Keratinocytes Using ROS Sensitive Fluorescent Sensors

Given the vital role that ROS production and oxidative stress play in tissue injury during the onset of OM [17], we further investigated the dynamic changes of ROS production in H_2_O_2_-treated OKF6 cells in vitro. By introducing the CM-H_2_DCFDA probe in keratinocytes, we could monitor any change in the ROS production state upon exposure to H-MW-HA, either in the absence, or the presence, of H_2_O_2_-induced oxidative stress by measuring the changes in CM-DCF fluorescence intensity under the fluorescent microscope. The OKF6 cells were treated with H_2_O_2_ (400 μM) for 4 h, the time point that, as shown in our previous experiments, coincided with the highest levels of ROS production for OKF6 when treated with H_2_O_2_.

As expected, and in agreement with the results obtained in measuring the ROS production and kinetics in OKF6, treatment of OKF6 cells with 400 μM H_2_O_2_ for 4 h resulted in increased ROS production (*p* < 0.0001) of 5.54 ± 1.37 folds increase, compared to the control group value (Figure 6A). In all examined fields, the increase in the fluorescent intensity of cells in the H_2_O_2_-treated group was apparent, compared to all other experimental groups, when the cells were examined under the fluorescent microscope (Figure 6B). Thus, we suggest that at the time point 4 h, the increased ROS production was attributed to treating OKF6 cells with H_2_O_2_, indicating that oxidative stress plays a part in cell injury.

### 3.8. Malondialdehyde (MDA) and Myeloperoxidase (MPO) Expression in a Mouse Model of Mucositis

To confirm the potential translatability of our findings from the in vitro study in a real-world scenario, we aimed to examine the redox status in an in vivo pre-clinical model of chemotherapy-induced mucositis. This was achieved through the utilization of our previously developed in vivo preclinical model which reproduced both the clinical and histologic features of mucositis in mice [52]. Specifically, we tested the levels of MDA (malondialdehyde) and MPO (myeloperoxidase) enzymes as a surrogate indicator of oxidative stress in the oral epithelia of mice treated with 5-FU and compared them with untreated control mice. MDA is a product of the peroxidation of cell membranes, a consequence of the overproduction of ROS caused by the release of MPO, an enzyme associated with inflammatory processes and predominantly found in neutrophils [58,59]. The immunoreactivity of MPO and MDA were quantified through immunohistochemical staining, as both MPO and MDA are oxidative stress biomarkers that have been previously determined to be relevant to the disease pathogenesis of chemotherapy-induced mucositis [60,61], with MPO being overexpressed in oral tissues of mice receiving 5-FU [53]. The results showed elevated MDA and MPO levels in the oral epithelia of the 5-FU treated mice, indicating increased oxidative stress in this group. Immunostaining experimental and controlled mouse sections showed MDA- and MPO-positive cell counts. MDA and MPO staining underwent significant increase in mice treated with 5-FU. Specifically, MDA immunostaining revealed a greater, statistically significant cell count in the experimental group, compared to the control (1383.33 ± 476.19 vs. 378.33 ± 104.62; *p* = 0.0234). MPO immunoreactivity was also confirmed to be significantly greater in the 5-FU treatment group than the control group (46.67 ± 14.84 vs. 237.33 ± 23.31; *p* = 0.0023), (Figure 7), in agreement with previous findings [53].

Overall, the study established a direct link between chemotherapy-induced mucositis and oxidative stress, with the excessive production of ROS being identified as an underlying mechanism.

## 4. Discussion

The molecular mechanisms underlying chemotherapy- and radiotherapy-induced oral mucositis involve the production of ROS at early stages and consequent activation of oxidative stress pathways [14,15,22,62]. We, therefore, wanted to establish a new experimental in vitro model of oxidative stress-induced oral mucositis in which cellular activities and new therapeutic interventions could be characterized. Given that the H_2_O_2_-induced oxidant model has been widely used to study cell response to oxidative stress [18], here, we developed a novel in vitro model of oxidative stress-induced oral mucositis using hydrogen peroxide to treat human oral epithelial cell lines. Specifically, in the present study, we studied the cytotoxic effect of H_2_O_2_ on proliferation, viability and apoptosis of oral keratinocytes and our results demonstrated that H_2_O_2_ led to an anti-growth effect on H400, H357, and OKF6 cells by both suppression of cell proliferation and induction of apoptosis. This in vitro model of H_2_O_2_-induced oral mucositis allowed the detection of changes in cellular activities in cultured oral keratinocytes, excluding other submucosal cellular components.

OM is a complicated pathological process that involves the effects of chemotherapy and radiotherapy on the keratinocytes and other cell types in the submucosal tissue, such as endothelial cells and fibroblasts [8,13,63]. To date, the exact mechanisms underlying the development of oral mucositis have not been completely elucidated [5,16,64]. Lack of a suitable in vitro model for oral mucositis could be a part of the possible difficulties in understanding the exact mechanism underlying the development of this condition. Monolayer cultures of oral keratinocytes have been widely used to mimic a specific part of the oral mucosal epithelial barrier [23,29]. Recently, human immortalized untransformed skin keratinocyte HaCaT line [65] has been more widely used to model the oral mucosa and for comparison with squamous head and neck cancer [66,67,68,69,70,71,72], whereas immortalized oral OKF6 keratinocytes [36] have been used less often to model oral mucositis [73]. In our model, we also used the human oral squamous cell carcinoma cell lines (H400 and H357), in analogy with the existing oral mucosa models [23]. Moreover, these cell lines are currently the best characterized oral epithelial cell lines available, easy to culture, and capable of generating highly reproducible results [35]. However, our results showed that the model can also be applied successfully using immortalized oral keratinocyte (OKF6) cell lines. Hence, this cell line was chosen for the successive experiments to model the oral mucosal epithelium, and for comparison with oral cancer cells.

In addition to cost and ethical considerations, an advantage of in vitro models over animal testing includes the ability to monitor dynamic and rapid mucosal biological responses to the oxidative stress damage. Recently, several models have been developed where the effects of either ionizing radiation or chemotherapeutic agents on oral keratinocyte proliferation were examined to reiterate the mucositis-associated epithelial changes observed in vivo [25,26,27,28]. By comparison, in our model we used H_2_O_2_ to induce the oxidative stress-induced cellular injury. Although we can use this model to ask focused mechanistic questions regarding the pathogenesis of oral mucositis, as well as the role of interventions to prevent the condition, it should be acknowledged that one of the limitations of this study is that monolayer cultures of oral keratinocytes represent two dimensional cultures so does not represent accurately what is occurring in situ, where the cells live in three dimensions [29]. Furthermore, oral mucositis is a condition where complex multi factors are included in its pathogenesis, which cannot be entirely optimized in vitro, including local and systemic factors (e.g., oral microbiota, xerostomia, neutropenia).

In the current study, the cytotoxic potential of H_2_O_2_ on oral keratinocyte cell lines was evaluated by means of trypan blue exclusion assay, fluorescence-based live–dead assay, and fluorometric cytotoxicity assay (FCA). Trypan blue exclusion is the most common test for cell viability; however, a number of limitations of this assay are related to the exclusion of certain dyes by live cell membranes. For example, cells must be counted within 3–5 min from the dye staining, because the number of blue-staining cells increases with time after the dye’s addition [74]. When counting a large number of samples, it is extremely inconvenient to perform all the tests on the same day by counting one cell suspension at a time before staining the next sample. Trypan blue exclusion is also a time- and cost-consuming assay. Therefore, a rapid and more reliable method of discriminating live cells from dead cells was required, and the FCA assay was chosen. The FCA belongs to a non-clonogenic microplate-based assays group that measures total living cell density after a short incubation time [75,76]. FCA applications are comparable to those of the commonly used colorimetric MTT assay [77], with slightly different endpoints. By measuring the fluorescence generated when the non-fluorescent probe fluorescein diacetate (FDA) is hydrolyzed, the FCA measures the esterase activity of cells with intact plasma membranes [76,78]. The fluorescence detection makes the FCA a more sensitive assay, when compared to the MTT assay [75]. The FCA has been used in a semiautomated 96-well setup since the late 1980s, and several publications have indicated its benefits in drug development using cancer cell lines [79,80,81]. Hence, it was deemed to be a reliable assay that was suitable for our experimental plan.

In our study, a broad range of concentrations of H_2_O_2_ (100–1200 μM) were tested in order to assess the toxic effects of continuous exposure to H_2_O_2_ on the viability and growth of oral keratinocyte cells. Our data showed that H_2_O_2_ was able to exert a dose- and time-dependent toxic effect on the tested epithelial cell lines. In fact, a significant change in cell survival was shown after incubation of H400 cells for 24 h with 400 μM H_2_O_2_ and higher. When exposed to different concentrations of H_2_O_2_ for 24 h, H400 cells, H357 cells, and OKF6 cells showed a significant dose-dependent decrease in cell survival. The IC_50_ value in the H400 cells, H357 cells, and OKF6 cells were approximately 800, 800, and 400 µM, respectively, based on FCA assays. These findings are consistent with other previous observations, indicating that H_2_O_2_ concentration-dependence and time-dependence promoted damage in different normal and malignant cells lines [22,82,83,84,85]. However, the approximate IC_50_ value of H_2_O_2_ in our study, after 24 h exposure, was higher than the IC_50_ (150 µM) reported by Zhou et al. using HTR-8/SVneo cells [84]. In a separate study, Moll et al. [85] used H_2_O_2_ concentrations of up to 1000 µM to evaluate apoptosis and proliferation in human term placentas. Furthermore, Murata et al. [86] used a concentration of 100 µM H_2_O_2_ for 24 h to determine apoptotic and invasion rates in term extra-villous trophoblast cell lines. The variability in results from these previous studies might be attributed to the variation in H_2_O_2_ sensitivity of the studied cells.

In a dose-dependent manner, H_2_O_2_ increased the number of dead cells, based on a trypan blue exclusion assay, and propidium iodide (PI)-positive cells in the H400 cells, suggesting that H_2_O_2_-induced epithelial cell death occurred via apoptosis. The H_2_O_2_-induced apoptotic cell death in the tested epithelial cell lines was evident from morphological studies (phase contrast and PI staining). Apoptosis of dying cells is characterized by particular morphological and biochemical alterations [87]. Changes include cell shrinkage, nuclear condensation and fragmentation, blebbing of the cell membrane, apoptotic body formation, in addition to the loss of cell attachment to neighbors, all of which are common features of apoptotic cells [88]. In this study, three different types of assays were performed to gain clues about the mode of cell death induced by H_2_O_2_ against H400, H357, and OKF6 cells: trypan blue exclusion, phase contrast microscopy, and fluorescence-based live–dead assay. Trypan blue exclusion is the most common test that can exclude live cells from non-viable cells [74]. Trypan blue exclusion study showed that H_2_O_2_ induced cell deathin H400 OSCC cells through a shift in live cell population proportional to increasing H_2_O_2_ concentrations from 2 to 24 h. One of the most cost-effective methods to determine cell apoptosis is morphological analysis through phase contrast and fluorescence inverted microscopy [89]. Our observations identified noticeable morphological changes consistent with apoptotic cell death in treated cells, including loss of cellular adhesion to neighbors, membrane blebbing, and chromatin condensation and fragmentation, along with the creation of apoptotic bodies. An FDA stain was used in conjunction with propidium iodide to determine dead cells after the loss of cell membrane integrity and the results showed nonviable cells were stained bright red.

Chemoradiotherapy-induced mucosal injury, particularly OM, is initiated by cytotoxic agents that cause oxidative stress in both mucosal and submucosal compartments [15,90,91]. Oxidative stress plays a key role in developing chemoradiotherapy-induced OM [15,92,93,94]. Among the key reactive oxygen species (ROS), hydrogen peroxide (H_2_O_2_) is an intermediate to both hypochlorite radical and hydroxyl radical (˙OH) production [95,96], and there is much evidence that H_2_O_2_ is a mediator of oxidative damage in cells [97]. Excessive production of these ROS affects several cellular structures and leads to DNA damage, protein modification, lipid peroxidation, disruption of cell signalling, and cellular death, causing destruction to extracellular matrix (ECM) components, such as collagen, hyaluronan, and proteoglycans [98,99], and further altering the metabolism of cells responsible for the synthesis of dermal ECM [100].

To better understand the cellular mechanism and the dynamics underlying this effect, we adopted a free radical (H_2_O_2_)-induced oxidative stress model that we established in vitro in our previous experiments. Here, we wanted to assess the redox status in oral keratinocyte cells when treated with H_2_O_2_. The CM-DCF-based fluorimetric assay revealed that H_2_O_2_ could induce ROS in human normal oral keratinocyte cells even after only 15 min of exposure, and the ROS levels increased in a time- and dose-dependent manner. Similar findings were reported in previous studies conducted with HaCaT human keratinocyte cells [101] and human umbilical vein endothelial cells (HUVECs) [102], although the exposure duration to H_2_O_2_ in those studies was typically longer than 1 h. Additionally, Cirillo et al. [22] demonstrated a significant increase in ROS levels in normal human oral fibroblast (NHOF) cells after transient exposure to 600 μM of H_2_O_2_ for more than 1 h over 5 days, which aligns with our results. More recent literature has shown that HaCaT cells exposed to 1 mM [103] or 1.6 mM [104] of H_2_O_2_ for less than 1 h also exhibited a significant increase in ROS levels, which further corroborates our findings. Overproduced ROS can cause damage to important biomolecules, such as DNA, proteins, and lipids [105]. For instance, hydrogen peroxide could cause oxidative DNA damage in normal human oral fibroblast (NHOF) cells [22]. Based on this evidence, we postulated that the balance between ROS generation and an antioxidant system was destroyed by the surplus ROS produced by H_2_O_2_, ultimately resulting in DNA damage. Further experiments, however, are needed to substantiate the oxidative DNA damage in normal human oral keratinocytes in a model of oxidative stress.

The in vivo arm of the study intended to further investigate the role of oxidative stress markers such as MDA and MPO in the pathogenesis of chemotherapy-induced OM using a murine model. Here, we reported a significantly higher number of cells stained with the antibodies of interest. This substantiates the current findings of the literature, which propose MDA and MPO as potential candidates for oxidative stress modulation, and link these stress markers to chemotherapy-induced mucositis [106,107].

The trend for MDA to be significantly higher in the chemotherapy-induced OM mouse models are complementary to the findings in the literature. It has been reported that, following exposure to antioxidants, murine models of OM showed lower MDA levels. Additionally, exposure to chemical and/or radiation insults, driving further oxidative stress, increase MDA levels [94]. Here, we reported a tendency for heightened MDA following 5-FU exposure, in isolation of other treatments, in a mouse model. MPO from our results appears promising as a potential therapeutic target in OM management. As an upstream molecule of the oxidative stress process, it is responsible for the lipid peroxidation that contributes to superoxide radical formation and, therefore, to oxidative stress [108]. The use of a well-established pre-clinical model and the evaluation of specific stress markers enhance the understanding of the pathophysiology of mucositis, contributing to potential future therapeutic strategies to mitigate its adverse effects during cancer treatment.

In summary, we successfully developed and characterised anin vitro model of oxidative stress-induced oral toxicity, using hydrogen peroxide, which is based on human oral epithelial cell lines. We evaluated the cytotoxicity and morphological changes induced by H_2_O_2_ on three different cell lines (H400, 357, and OKF6) and found that H_2_O_2_ inhibited cell proliferation and induced marked morphological changes. Furthermore, our study showed that H_2_O_2_ induced a dose-dependent cell death that was morphologically compatible with apoptosis and increased ROS levels, leading to damage in important biomolecules, such as DNA, proteins, and lipids. To better understand the toxicological effect of H_2_O_2_ on epithelial cells, future research should focus on understanding the molecular markers and signaling pathways involved in cell apoptosis, such as assessing caspase activity and P43 protein expression, via cell-based assays and western blot analysis. Additionally, we used a previously developed murine-model of chemotherapy-induced mucositis to investigate and quantify levels of MDA and to confirm MPO upregulation. MDA and MPO are key markers of oxidative stress and their correlations with oral mucositis in 5-FU treated mice corroborates the role of oxidative stress in the pathogenesis of mucositis. Our findings support previous studies that suggest these markers may be potential targets for oxidative stress modulation to mitigate the effects of chemoradiation in the oral mucosa [109,110].

## 5. Conclusions

Our study successfully established an in vitro model of H_2_O_2_-induced oral toxicity using human oral epithelial cell lines. This model detected changes in cellular activities in cultured oral keratinocytes, excluding other submucosal cellular components, and demonstrated the dose-dependenteffect of H_2_O_2_ on the proliferation, viability, and production of ROS in the studied cell lines. Our findings suggest that this model is a suitable tool for understanding the pathogenesis of oral mucositis and for evaluating potential interventions to prevent the condition. Furthermore, the in vivo arm of the study revealed that MPO and MDA may be significant markers in the oxidative stress pathway associated with OM and could potentially be targeted therapeutically. These valuable insights can guide future research into developing effective interventions to alleviate the deleterious effects of anti-cancer therapy.

## Figures and Tables

**Figure 1 biomolecules-13-01239-f001:**
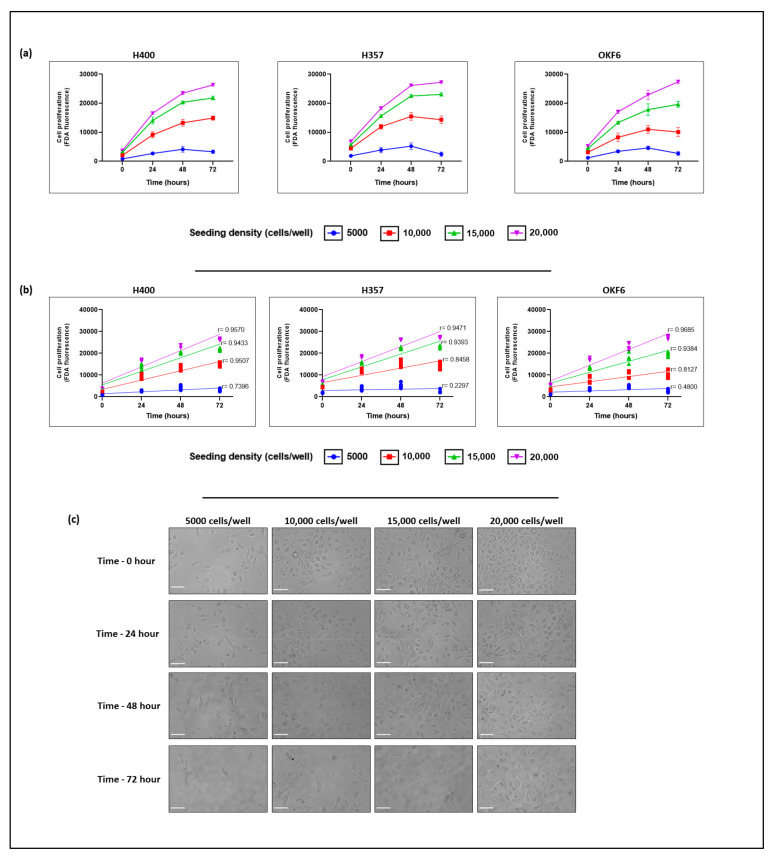
The proliferation of H400, H357, and OKF6 cells over 72 h at varying cell seeding densities. Three monolayer cultures of oral mucosa keratinocyte (OSCC cell lines, H400 and H357, immortalized normal human oral keratinocytes, OKF6) seeded in 96-well plates at 5 × 10^3^, 1 × 10^4^, 1.5 × 10^4^, and 2 × 10^4^ cells/well were grown under standard conditions (37 °C, in a 5% CO_2_ incubator). Cells were observed over 72 h, replenishing culture medium every 24 h. (**a**) Cell proliferation and viability at four different time points (0, 24, 48, and 72 h) assessed by fluorescein diacetate (FDA) fluorescence in Synergy HTX Multi-Mode Reader (Bio-Tek, USA) at the respective excitation and emission wavelengths of 485/20 nm and 528/20 nm. (**b**) Pearson correlation index between viable oral keratinocyte cell number and fluorescence intensity over time. (**c**) Representative micrographs of OKF6 cells showing confluence over time. Images (FLoid cell imaging system, Life Technologies) with magnification at 20×. Scale bar of 100 µm applies to all panels. All measurements were performed in triplicate with mean ± SD for 5 technical replicates per cell line.

**Figure 2 biomolecules-13-01239-f002:**
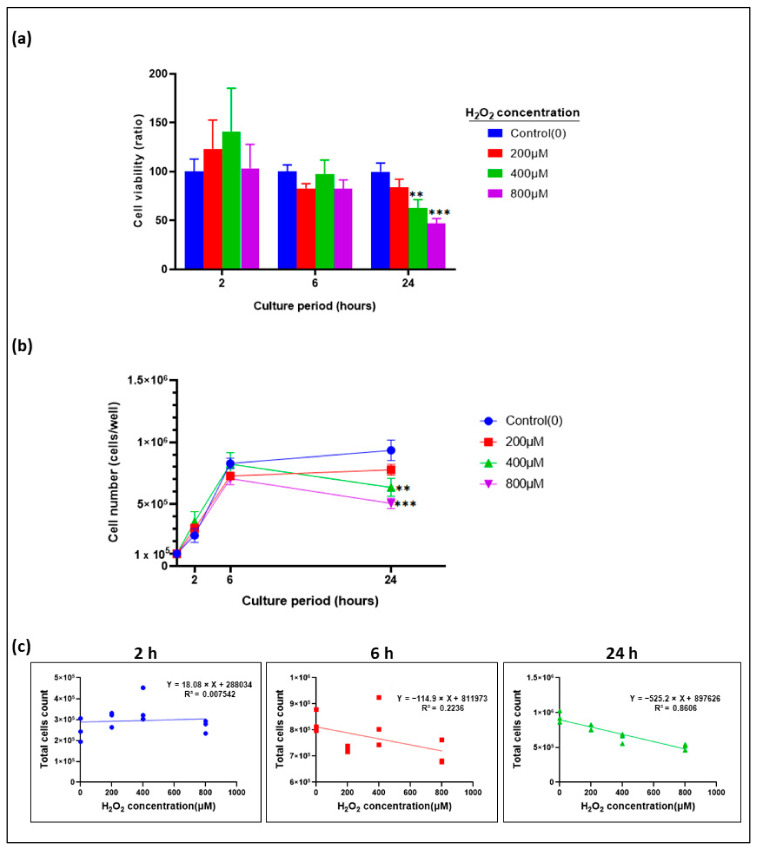
Effect of H_2_O_2_ treatment on the H400 cell growth. (**a**) H400 cells were exposed to increasing concentrations of H_2_O_2_ for the indicated times. Cell viability was measured at time points by using trypan blue exclusion assay using a TC10™ Automated Cell Counter (Bio-Rad). Bars indicate mean cell viability ratios in H_2_O_2_-treated over untreated cells. Error bars represent one standard deviation of the mean of three technical replicates. Statistical significance is given as follows: ** *p* < 0.005 and *** *p* < 0.001 as compared to control. (**b**) H400 cultures were treated with increasing concentrations of H_2_O_2_ for the indicated times and cell count was measured by using the TC10™ Automated Cell Counter (Bio-Rad). Data are represented as mean cell number ± SD of three technical replicates. Statistical significance is given as follows: ** *p* < 0.005 and *** *p* < 0.001 as compared to control. (**c**): Pearson correlation index between total H400 cells number and H_2_O_2_ dose over time.

**Figure 3 biomolecules-13-01239-f003:**
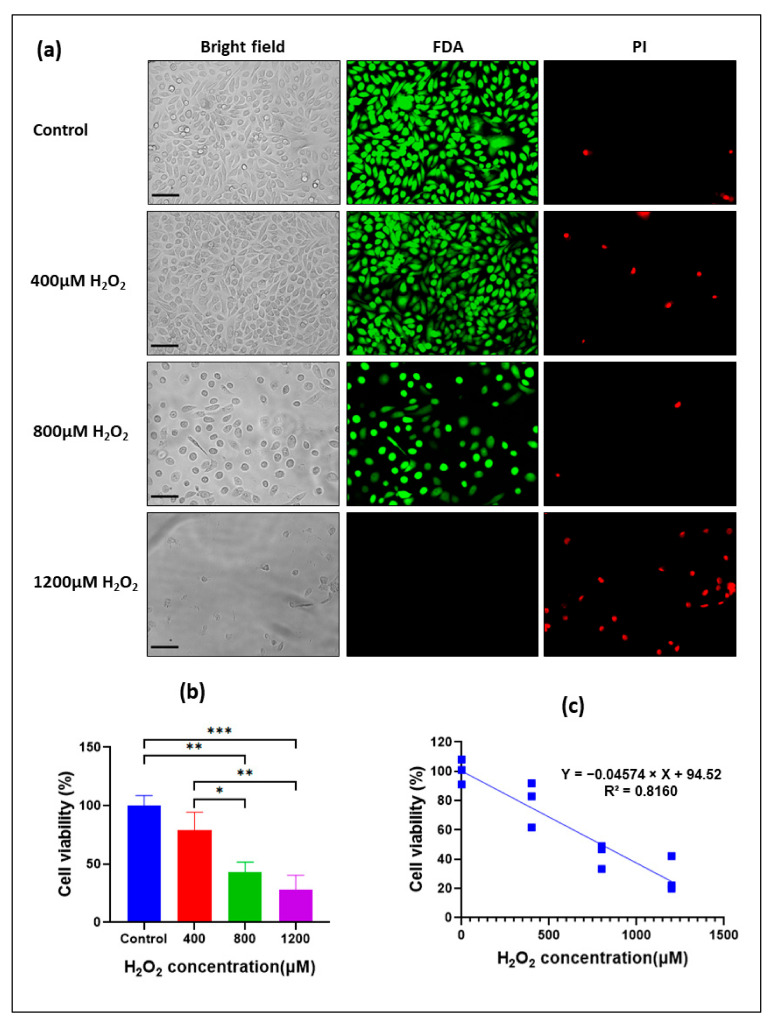
H_2_O_2_-induced cell death in oral keratinocytes. Representative fluorescein diacetate (FDA) and propidium iodide (PI) staining of H400 cells exposed to increasing concentrations of H_2_O_2_ for 24 h. Cell viability was quantified at the 24 h time point using an inverted fluorescence microscope (FLoid™ Cell Imaging Station, Life Technologies Australia). Fluorescence of vital cells was measured in six different random fields/well per treatment in three technical replicates. Captured images were transferred to a computer and analyzed to count stained vital cells using Image J Software (ImageJ v. 1.50i, National Institutes of Health). (**a**) Representative micrographs (20×) of H400 cells treated with H_2_O_2_ showed vitality staining of the cells (from left to right: Bright-field, FDA signal, and PI signal). The same field is shown in each image filter condition. Scale bar represents 100 μm and applies to all panels. (**b**) Mean cell viability ratios in H_2_O_2_-treated over untreated H400 cells. Error bars represent one standard deviation of the mean of three technical replicates. (**c**) Pearson correlation index between cell viability ratio and H_2_O_2_ dose after 24 h of H_2_O_2_ incubation. Data are expressed as mean ± SD of three technical replicates. * *p* < 0.05, ** *p* < 0.005 and *** *p* < 0.001.

**Figure 4 biomolecules-13-01239-f004:**
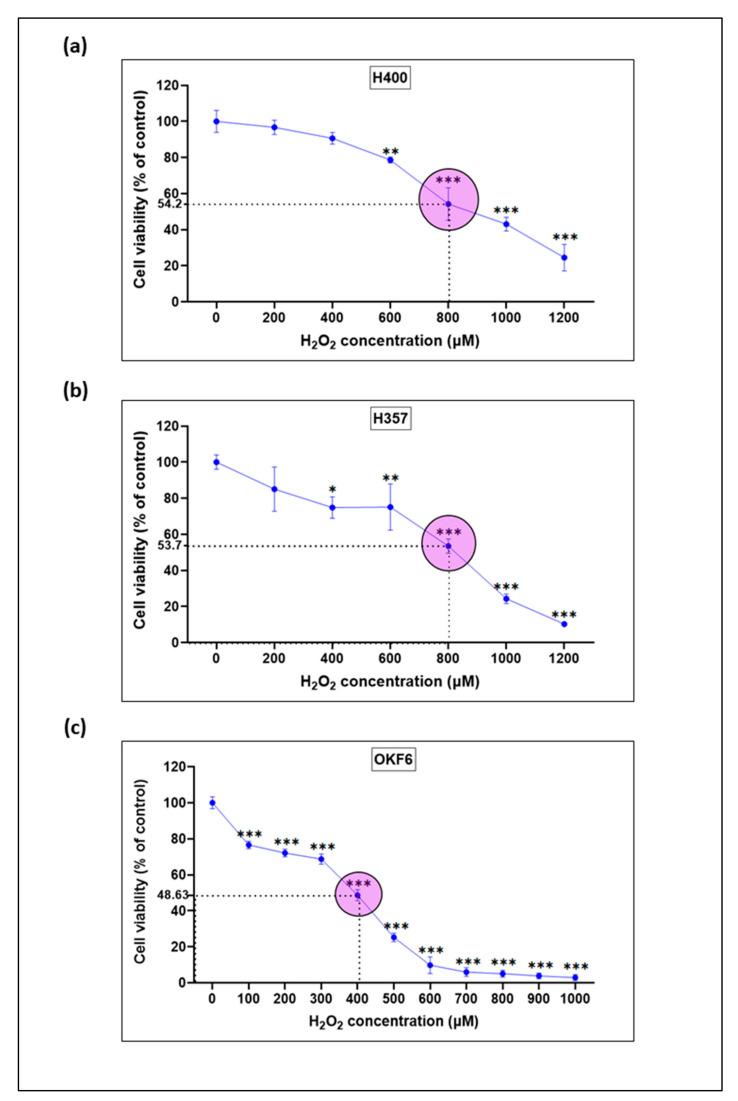
H_2_O_2_-induced loss of cell viability in malignant and normal oral epithelial cells. H400 (**a**), H357 (**b**), and OKF6 (**c**) cells, were seeded at densities of 1.5 × 10^4^ in 96-well plates and incubated for 24 h with increasing concentrations of H_2_O_2_. Cell viability (**a**–**c**) after 24 h of incubation with H_2_O_2_ was measured by fluorescein diacetate (FDA) fluorescence in a Synergy HTX Multi-Mode Reader (Bio-Tek, USA). The data are expressed as the relative response of treated cells as compared to untreated controls (100%) and represent means ± SD from at least three technical replicates of 1–3 biological replicate experiments. Statistical significance is given as follows: * *p* < 0.05, ** *p* < 0.005, and *** *p* < 0.005, as compared to untreated controls.

**Figure 5 biomolecules-13-01239-f005:**
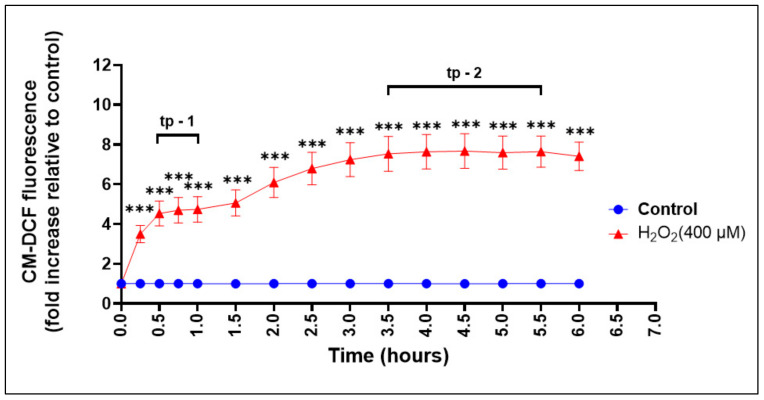
Effect of H_2_O_2_ (400 μM) on the induction of ROS production. OKF-6 cells were cultured in complete keratinocyte serum free media (K-SFM, #17005-042, Thermo Fisher Scientific) for 24 h in 96-well culture plates (Cat#: CLS3603; Sigma-Aldrich) under standard conditions (5% CO2 at 37 °C). Culture medium was then removed and cells were incubated with the fluorescent probe, 5-(and-6)-chloromethyl-2′,7′-dichlorodihydrofluorescein diacetate (CM-H_2_DCFDA), for 30 min. A fluorescent probe was then replaced with complete culture medium in the absence, or presence, of H_2_O_2_ (400 μM). ROS production was measured as the change in CM-DCF fluorescence (-fold increase) relative to control as CRT = 1 at timepoints 0, 15 min, 30 min, 45 min, and every 30 min for up to 6 hrs. CM-DCF fluorescence intensity was measured using Synergy HTX Multi-Mode Reader (Bio-Tek, USA) with maximum excitation and emission spectra of 495 nm and 529 nm, respectively. Data are from one independent experiment, n = 4 per group (four technical replicates). Values represent mean ± SD, where *** *p*< 0.001, compared to the H_2_O_2_ (400 μM) treated group. Abbreviations: tp-1, time plateau-1; tp-2, time plateau-2.

**Figure 6 biomolecules-13-01239-f006:**
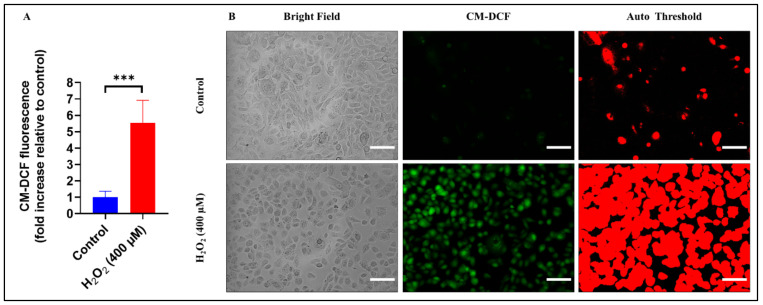
H_2_O_2_-induced ROS production in oral keratinocytes. OKF-6 cells were cultured in complete keratinocyte serum free media (K-SFM, #17005-042, Thermo Fisher Scientific) for 24 h in 6-well culture plates (Cat#: CLS3516; Sigma-Aldrich) under standard conditions (5% CO_2_ at 37 °C). Culture medium was then replaced with complete culture medium supplemented either in the absence or the presence of H_2_O_2_ (400 μM) and the cells were incubated for 4 h. Culture medium was then removed and the cells were incubated with the fluorescent probe, 5-(and-6)-chloromethyl-2′,7′-dichlorodihydrofluorescein diacetate (CM-H_2_DCFDA), for 30 min. Then, the fluorescent probe was replaced with Hank’s buffer saline solution (HBSS), and the plates were viewed under the FLoid^TM^ cell imaging station digital inverted microscope (ThermoFisher Scientific, USA). The microscope settings selected included the fluorescein (FITC) filter (488 nm excitation and 530 nm emission). (**A**) The intensity of CM-DCF fluorescence was measured in seven different fields per well per condition (seven technical replicates) in three independent experiments (three biological replicates). ROS production was measured as the change in CM-DCF fluorescence (-fold increase) relative to control as CRT = 1. Values represent mean ± SD. Where, *** *p* < 0.001. (**B**) Representative micrographs, at 20× magnifications, of oral keratinocytes treated with, or without, H_2_O_2_ (400 μM). Fluorescence intensities of oxidized probes (CM-DCF from CM-H2DCF) were quantified using the ImageJ software. Image threshold was set using Auto-threshold method (Huang) and the mean intensity of the threshold area was measured. The representative micrographs showed the bright field (left panel), fluorescent intensity of 5-(and-6)-chloromethyl-2′,7′-dichlorofluorescein (CM-DCF) (middle panel), and Auto threshold (right panel). The same field is shown in each image filter. Scale bar represents 100 μm and applies to all panels.

**Figure 7 biomolecules-13-01239-f007:**
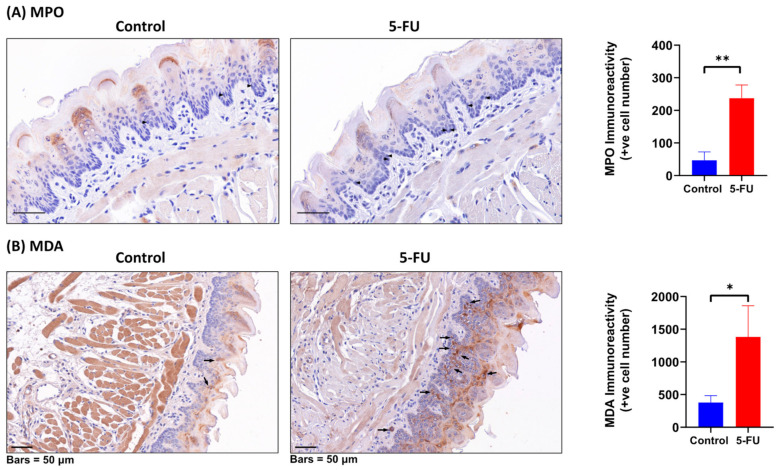
MPO and MDA immunostaining quantification and analyses. Control and 5-FU treated mice tongue samples were immunostained with DAB for MPO (**A**) and MDA (**B**), and counterstained with hematoxylin. Highlighted here, by the black arrows, are cells positively stained with DAB (reacting to MPO or MDA positive antigens) and hematoxylin. MPO and MDA immunoreactivity were quantified and represented graphically. Values represent mean ± SD. Where, ** *p* < 0.01, and * *p* < 0.05. MPO, Myeloperoxidase; MDA, Malondialdehyde; DAB, Diaminobenzidine.

## Data Availability

Full datasets are available upon reasonable request to the corresponding author.

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
