# Peer review of "Assessment of Oxidative Stress-Induced Oral Epithelial Toxicity"

_biomolecules, 2023, doi:10.3390/biom13081239_

Round 1

Reviewer 1 Report

This manuscript is an original basic-science research aimed at establishing an in vitro model of oxidative stress-induced oral mucosal toxicity.

The study was performed with high methodological standards, making results scientifically sound and strong. Conclusions are very interesting, and the clinical implications are very promising.

This piece of science deserves a place in the literature as it truly represents the notion of translational research in the field of oral medicine and medicine in general. Authors need to be congratulated for their hard work!

Author Response

The authors gratefully thank this referee for their appreciation of our work!

Reviewer 2 Report

In this paper, the authors have characterized and validated a model for oxidative stress in human oral keratinocyte cells and have succeeded in finding optimal experimental conditions for the study of oxidative stress.

The paper is well written and contains detailed explanations, especially on materials and methods. 

But it seems that the part about the mouse model was subsequently added to the work. It does fit meaningfully, but I think it should be better incorporated . Please, add some sentences with additional explanations and the purpose of this experiment.

In chapter 2.6 you describe in detail the method to perform the experiment, but I do not see the point. You use the term 5- FU, as an abbreviation, and there is no explanation of what it is. You relied a bit too much on your previous work there.

I like how you explain the initial increase in proliferation and subsequent decrease in cell number (bottom of page 8). Are there similar examples of this behavior by other authors?

Please match the text describing the statistical significance in Figure 7 with the other images.

In Figure 7 you have exactly the same picture/graph of MPO control as in your earlier paper. I don't know if this is admissible or actually allowed.

Author Response

Many thanks for your comments. Please see attached response. 

Reviewer 3 Report

Thank you for your paper. It is very interesting, the work is well conducted and results interesting. Language is good. Thank you again for your paper

Author Response

(The authors gave the same response as above.)

Reviewer 4 Report

This manuscript focuses on the influence of oxidative stress in oral toxicity on the epithelial tissue. The topic is not novel, several works are in literature.

The aim of the authors was the introduction of an invitro model, however the number of experiments is poor. They tested different concentration of H2O2 on different cell lines. 

They evalueted the viability and proliferation of cells using three different methods, however the aim remains the saim. I find these methods and results redundant.

The authors used a methods based on the use of microscopy, but in the methods they wrote Real Time. The Real time requires the extraction of nucleic acids and then their amplification. Therefore, the authors used the terms real time in inappropriate manner.

In the Fig.3a the scale bar is in the caption, but not in the images. I suggest adding it in the single figure.

A check of English is required. For example, the sentence at the lines 72-76 is not clear. 

Author Response

Thanks for your comments - response attached. 

Reviewer 5 Report

the study by Mohammed et colleagues is well conducted and the conclusion were supported by the results. However, the author should be included in the Introduction section 3 recent papers that investigated oral mucositis and ROS involvement in this disease.

1) Picciolo G, Mannino F, Irrera N, Minutoli L, Altavilla D, Vaccaro M, Oteri G, Squadrito F, Pallio G. Reduction of oxidative stress blunts the NLRP3 inflammatory cascade in LPS stimulated human gingival fibroblasts and oral mucosal epithelial cells. Biomed Pharmacother. 2022 Feb;146:112525. doi: 10.1016/j.biopha.2021.112525. 

2)Picciolo G, Mannino F, Irrera N, Altavilla D, Minutoli L, Vaccaro M, Arcoraci V, Squadrito V, Picciolo G, Squadrito F, Pallio G. PDRN, a natural bioactive compound, blunts inflammation and positively reprograms healing genes in an "in vitro" model of oral mucositis. Biomed Pharmacother. 2021 Jun;138:111538. doi: 10.1016/j.biopha.2021.111538. 

3)Picciolo G, Pallio G, Altavilla D, Vaccaro M, Oteri G, Irrera N, Squadrito F. β-Caryophyllene Reduces the Inflammatory Phenotype of Periodontal Cells by Targeting CB2 Receptors. Biomedicines. 2020 Jun 17;8(6):164. doi: 10.3390/biomedicines8060164. 

Minor editing of English language are required

Author Response

Thank you for your valuable feedback. We have now included the three suggested studies in our introduction section. On page 2, lines 83-84

Round 2

Reviewer 4 Report

 Due to high Journal level, I do expect a more complete study, exploring more cell experiments since just a few parameters were varied. I don’t find useful using several methods to test the same aspect, the proliferation. To validate an in vitro model other experiments would be needed.

In addition, the authors in the revised version did not improve the manuscript.